# Firm Survival and Gender of Firm Owner in Times of COVID-19: Evidence from 10 European Countries

Joachim Wagner [1,2]

1  Kiel Institute for the World Economy, Leuphana University of Lüneburg, 21314 Lüneburg, Germany; wagner@leuphana.de
2  IZA Institute of Labour Economics, 53113 Bonn, Germany

**Abstract:** This paper uses firm level data from World Bank Enterprise surveys conducted in 2019, and COVID-19 follow-up surveys conducted in 2020, in ten European countries to investigate the link between the gender of the firm's owner and the firm's survival until 2020. The empirical investigation uses econometric models that control for the firm's characteristics that are known to be related to firm survival. The estimated effect of female ownership is positive ceteris paribus. Furthermore, the size of this estimated effect can be considered to be large on average. Having a female owner helped firms to survive.

**Keywords:** gender; female owned firms; firm survival; COVID-19; World Bank Enterprise surveys

**JEL Classification:** D22; L20; L25; L29

## 1. Motivation

When the coronavirus and COVID-19 reached Europe in the first quarter of 2020, firms were hit by negative demand shocks due to quarantine and lockdown measures. Furthermore, supply chains were damaged and this led to negative supply shocks. These shocks had a negative impact on many dimensions of firm performance. Waldkirch (2021) reports evidence on the impact of the COVID-19 pandemic on firms around the world based on the so-called COVID-19 follow-up surveys to the World Bank's Enterprise Surveys conducted in 2020.

Some firms were hit so hard by these negative exogenous shocks that they decided to close down permanently. Waldkirch (2021, p. 6) reports that in samples of firms, collected in 21 countries in Europe, Latin America, Africa, and Asia in 2019/2020, 4.1 percent of the businesses had permanently closed. In the sample of firms used here in this study, the share of exits is 4.59 percent (see Table 1).

An important question is which characteristics of firms help many of them to survive the pandemic. In addition to the usual suspects, which have been discussed at length in the literature over recent decades, and include firm demographics such as firm age, firm size, exports, productivity, and innovation (and that will be looked at in more detail in Section 2 of this paper), one firm characteristic that has, to the best of my knowledge, not been considered yet in the context of the COVID-19 pandemic, is the gender of the firm's owner. In their comprehensive survey of the literature, Jennings and Brush (2013, p. 671) argue that many empirical studies find that with respect to many standard economic indicators, female-led firms tend to exhibit inferior performance; however, Jennings and Brush (2013, p. 672) point out that a small yet growing number of studies challenge whether firms that are female-owned invariably underperform relative to male-owned firms. There are studies that reveal mixed results, no significant differences, or even an advantage to female-led firms. "In terms of survival, for instance, the evidence is mixed." (Jennings and Brush 2013, p. 672).

**Table 1.** Descriptive evidence on share of exits by gender of owner in 10 European countries, 2019/2020.

| Country | No. of Firms | Share of Exits (Percent) | Share of Female Owned Firms (Percent) | Share of Exits among Female Owned Firms (Percent) | Share of Exits among Male Owned Firms (Percent) |
|---|---|---|---|---|---|
| All countries | 6013 | 4.59 | 36.37 | 3.98 | 4.94 |
| Bulgaria | 552 | 6.70 | 37.86 | 5.74 | 7.29 |
| Croatia | 349 | 2.58 | 32.09 | 0.89 | 3.38 |
| Czech Rep. | 398 | 1.76 | 32.91 | 1.53 | 1.87 |
| Hungary | 623 | 1.77 | 49.12 | 0.98 | 2.52 |
| Italy | 439 | 7.74 | 20.96 | 3.26 | 8.93 |
| Poland | 887 | 2.93 | 39.80 | 3.68 | 2.43 |
| Portugal | 795 | 9.69 | 43.65 | 7.49 | 11.38 |
| Romania | 518 | 3.47 | 36.68 | 5.26 | 2.44 |
| Russia | 1116 | 3.94 | 29.66 | 4.83 | 3.57 |
| Slovak Rep. | 336 | 3.87 | 34.52 | 0.86 | 5.45 |

Source: Own calculations based on the World Bank Enterprise surveys; for details, see text.

Although we have (to the best of my knowledge) no empirical evidence on the difference of firm survival by gender of firm owner during the COVID-19 pandemic, one may hypothesize that female owned firms have a lower rate of closure because women tend to be more risk adverse than men, and therefore, firms lead by men might be engaged in more risky businesses and have a higher risk of failure, especially in times of negative demand and supply shocks.

This paper contributes to the literature by using firm level data from ten European countries collected in the World Bank's Enterprise Surveys in 2019, and from the COVID-19 follow-up surveys conducted in 2020, to investigate the link between gender of the firm's owner and firm survival, controlling for other determinants of firm exit.

The rest of the paper is organized as follows: Section 2 introduces the data used and discusses the variables that are included in the empirical model to test for the role of the gender of the firm's owner in firm survival. Section 3 reports descriptive evidence and results from the econometric investigation. Section 4 concludes.

## 2. Data and Discussion of Variables

The firm level data used in this study are taken from the World Bank's Enterprise Surveys in 2019 and from the COVID-19 follow-up surveys conducted in 2020.[1] These surveys were conducted in a large number of countries all over the world. In this study we focus on countries from Europe. All countries with complete data for at least five firms, that took part in the 2019 survey, and reported in the 2020 follow-up survey that they had permanently closed down, are included in the study. This leaves us with data for ten countries: Bulgaria, Croatia, Czech Republic, Hungary, Italy, Poland, Portugal, Romania, Russia, and the Slovak Republic.[2]

The classification of firms as survivors or exits is based on question B.0[3] in the follow-up survey from 2020. Firms that participated both in the regular 2019 survey and in the follow-up survey were asked "[c]urrently is this establishment open, temporarily closed (suspended services or production), or permanently closed?" Firms that answered "permanently closed" are classified as exits, the other firms are considered to be survivors.

The classification of firms into female-owned firms and male-owned firms is based on the answer to question B4 in the regular survey of 2019. When there are females amongst the owners of the firm, a firm is considered to be a female-owned firm (and a male-owned firm otherwise).

Descriptive evidence on the share of firm exits by gender of the firms' owners in the total sample and by country is reported in in Table 1. Although the overall share of exits among female owned firms is 3.98 percent, it is about one percentage point higher among male-owned firms (4.94 percentage points). This raw difference in favor of female-owned firms can be considered large. Results differ by country, but in seven out of ten countries, the exit rate is smaller among female-owned firms.

In the empirical investigation of the link between the gender of the firm's owner and firm survival, a number of firm characteristics that are known to be correlated with firm exit (and that might be related to the gender of the firm's owner as well) are controlled for.

Their link to firm survival, and the way they are measured here, is discussed below (see Wagner 2021).

*Firm size*: Audretsch (1995, p. 149) mentions as a stylized fact from many empirical studies on exits that the likelihood of a firm exit apparently declines with firm size (usually measured by the number of employees in a firm). This is theoretically linked to the hypothesis of "liability of smallness" from organizational ecology. A small size can be interpreted as a proxy variable for a number of unobserved firm characteristics, including disadvantages of scale, higher restrictions on the capital market leading to a higher risk of insolvency and illiquidity, disadvantages of small firms in the competition for highly qualified employees, and lower talent of management (Strotmann 2007). For Germany, Fackler et al. (2013) show that the mortality risk falls with establishment size, which confirms the liability of smallness.

Firm size is measured as the number of permanent, full-time individuals that worked in the establishment at the end of the last complete fiscal year at the time of the regular 2019 enterprise survey (see question I.1).

*Firm age*: Audretsch (1995, p. 149) mentions as another stylized fact from many empirical studies on exits that the likelihood of a firm exit apparently declines with firm age, too. This positive link between firm age and probability of survival is labelled "liability of newness" and it is related to the fact that older firms are "better" because they spent a longer time in the market, during which they learned how to solve the range of problems facing them in day-to-day business. For Germany, Fackler et al. (2013) find that the probability of exit is substantially higher for young establishments which are not more than five years old, thus confirming the liability of newness.

Firm age is measured as follows. In question B.5 of the regular survey in 2019, firms were asked "[i]n what year did this establishment begin operation?". Firm age is the difference between 2019 and the founding year.

*Exports*: Exporting can be considered as a form of risk diversification through spread of sales over different markets with different business cycle conditions or in a different phase of the product cycle; therefore, exports might provide a chance to substitute sales at home by sales abroad when a negative demand shock hits the home market and would force a firm to close down otherwise (see Wagner 2013). Furthermore, Baldwin and Yan (2011, p. 135) argue that non-exporters are, in general, less efficient than exporters (younger, smaller, and less productive) and that, as a result, one expects that non-exporters are more likely to fail than exporters.

A number of recent empirical studies look at the role of international trade activities in shaping the chances for survival of firms; Wagner (2012, p. 256ff.) summarizes this literature. As a rule, the estimated chance of survival is higher for exporters, and this holds after controlling for firm characteristics that are positively associated with both exports and survival (such as firm size and firm age). This might point to a direct positive effect of exporting on survival.

The firm is considered to be an exporter if it reports any direct exports in question D.3 of the regular enterprise survey in 2019.

*Productivity*: In theoretical models for the dynamics of industries with heterogeneous firms, productivity differentials play a central role for entry, growth, and exit of firms. In terms of the equilibrium growing and shrinking, exiting and entering firms that have different productivities are found in an industry. These models lead to hypotheses that can be tested empirically. Hopenhayn (1992) considers a long-run equilibrium in an industry with many price-taking firms producing a homogeneous good. Output is a function of inputs, and a random variable that models a firm specific productivity shock. These shocks are independent between firms and are the reason for the heterogeneity of firms. There are sunk costs to be paid at entry, and entrants do not know their specific shock in advance. Incumbents can choose between exiting or staying in the market. When firms realize their productivity shock, they decide the profit maximizing volume of production. The model assumes that a higher shock in $t + 1$ has a higher probability the higher the shock is in $t$. In

equilibrium, firms will exit if, for given prices of output and input, the productivity shock is smaller than a critical value, and production is no longer profitable.

Farinas and Ruano (2005, p. 507f.) argue that this model leads to the following testable hypothesis: firms that exit in year $t$ were $t - 1$ less productive than firms that continue to produce in $t$. They test this hypothesis using panel data for Spanish firms. The hypothesis is supported by the data. Wagner (2009) replicates the study by Farinas and Ruano (2005) with panel data for West and East German firms from manufacturing industries. For the cohorts of exit from 1997 to 2002, the results are in line with the results for Spain.

Unfortunately, however, there is no suitable measure of productivity in the World Bank Enterprise survey, so productivity cannot be controlled for in the empirical models that test for a link between the gender of the firm's owner and firm survival; however, productivity is controlled for indirectly by including the information on the exporter status of the firm, because it is a stylized fact that has been found in hundreds of empirical studies from countries all over the world that exporters tend to be much more productive than non-exporters from the same narrowly defined industry (see Wagner 2007 for a survey).

*Foreign ownership*: Baldwin and Yan (2011) argue that from a theoretical point of view, the relationship that should be expected between foreign ownership and firm exit is not clear. On the one hand, foreign owned firms may have access to superior technologies belonging to their foreign owners that might increase their efficiency and lower the risk of exit. Their greater propensity to invest in R&D might lead to more innovations, improve their competitiveness at home and in foreign markets, and might therefore increase the chance of survival. On the other hand, Baldwin and Yan (2011) point out that foreign owned firms are less rooted in the host country economy, and they can shift their activities to another country when the local economy deteriorates. This should increase the probability of shutdown compared with nationally owned firms.

With a view on the COVID-19 pandemic, Waldkirch (2021, p. 4) argues that "on the one hand, multinational companies may be better able to weather the storm, as they are more financially stable or have access to multiple sources of inputs, thereby minimizing disruptions to the supply chain. On the other hand, these firms may also be exposed to the pandemic's impacts on a larger scale, in multiple countries, and at different times given the differential timing of the virus's spread and mandated quarantines and shutdowns in different countries".

A number of recent micro-econometric studies use firm level data for foreign owned firms and domestically controlled firms to investigate the (ceteris paribus) relationship between foreign ownership and firm survival. The Wagner and Weche Gelübcke (2012) survey of 26, mainly country specific studies, uses data from 17 developed and developing countries, two of which use data on affiliates worldwide. The big picture emerging from the findings of these studies can be summarized as follows. Results are highly country dependent. Foreign affiliates were found to be more likely to exit as compared with their domestic counterparts in Ireland, Belgium, Spain, and Indonesia, but less likely to exit in Canada, Italy, Taiwan, and the US. No significant differences in closure rates due to foreign ownership could be revealed for Japan, Turkey, and the UK.

In the regular survey in 2019, firms were asked what percentage of this firm is owned by private foreign individuals, companies, or organizations (see question B2). Firms that reported a positive amount here are considered as (partly) foreign owned firms.

*Innovation*: Josef Schumpeter (1942, p. 84) argued some 80 years ago that innovation plays a key role for the survival of firms, because it "strikes not at the margins of the profits and the outputs of the existing firms but at their foundations and their very lives". Baumol (2002, p. 1) called innovative activity "a life-and-death matter for the firm". This positive link between innovation and firm survival is found in a number of empirical studies. For example, Cefis and Marsili (2005) show that firms benefit from an innovation premium in that ceteris paribus extends their life expectancy; process innovation in particular seems to have a positive effect on firm survival.

In the regular survey in 2019, firms were asked whether during the last three years, this establishment has introduced new of improved products and services (see question H1). Firms that answered in the affirmative are considered to be product innovators. Similarly, firms were asked whether during the last three years, this establishment introduced any new or improved process, including methods of manufacturing products or offering services; logistics, delivery, or distribution methods for inputs, products, or services; or supporting activities for processes (see question H5). Firms that answered in the affirmative are considered as process innovators.

*Web presence*, i.e., having a website where potential customers can learn about, and order, goods or services, when personal contacts are not possible due to quarantine, and lockdown is often mentioned in the business press as a factor that might help firms to survive in the pandemic. Wagner (2021) uses the same data used here to test this hypothesis. He finds that the estimated effect of web presence on firm survival is positive, statistically significant, and large.

In the regular 2019 survey, firms were asked in question C22b, "[a]t present time, does this establishment have its own website or social media page?" Firms that answered "yes" are classified as a firm with web presence.

Furthermore, firms are divided by broad sectors of activity (manufacturing, retail/wholesale, construction, hotel/restaurant, and services) based on their answer to the question for the establishment's main activity and product, measured by the largest proportion of annual sales (see question D1a1).

Descriptive information on the difference between female and male owned firms in the sample are reported in Table 2. Compared with male owned firms, female owned firms have a slightly lower web presence, are smaller, are product innovators more often, are foreign owned firms less often, and are exporters less often. Furthermore, they are more often in the retail/wholesale and hotel/restaurant industry, and less often in construction and services.

**Table 2.** Difference between female and male owned firms in the sample table.

| Variable | Female Owned Firms | Male Owned Firms |
|---|---|---|
| Web-presence (Dummy; 1 = yes) | 0.6936 (0.46) | 0.7357 (0.44) |
| Firm age (Years) | 21.84 (16.53) | 20.60 (14.50) |
| Firm size (Number of employees) | 62.75 (135.69) | 92.43 (432.36) |
| Product innovator (Dummy; 1 = yes) | 0.2437 (0.42) | 0.2062 (0.40) |
| Process innovator (Dummy; 1 = yes) | 0.1280 (0.33) | 0.1270 (0.33) |
| Foreign owned firm (Dummy; 1 = yes) | 0.0453 (0.21) | 0.0844 (0.28) |
| Exporter (Dummy; 1 = yes) | 0.2515 (0.43) | 0.2882 (0.44) |
| Manufacturing (Dummy; 1 = yes) | 0.6255 (0.48) | 0.6349 (0.48) |
| Retail/Wholesale (Dummy; 1 = yes) | 0.2318 (0.42) | 0.1832 (0.39) |
| Construction (Dummy; 1 = yes) | 0.0366 (0.19) | 0.0659 (0.25) |
| Hotel/Restaurant (Dummy; 1 = yes) | 0.0425 (0.20) | 0.0311 (0.17) |
| Services (Dummy; 1 = yes) | 0.0636 (0.24) | 0.0849 (0.28) |
| Number of firms | 2187 | 3826 |

The table reports the mean values of the variables used in the estimation by gender of firm owner (standard deviations in brackets).

Descriptive statistics for all variables used in the empirical investigation are reported for the whole sample in the Appendix A Table A1.

### 3. Testing for the Role of Gender of the Firm's Owner in Firm Survival

To test for the role of the gender of the firm's owner in firm survival, empirical models are estimated with an indicator variable for firm survival, or not, until 2000 as the endogenous variable, an indicator variable for female-owned firms, or not, in 2019 as the exogenous variable, and various sets of control variables. All models are estimated by Probit, and average marginal effects with prob-values to indicate their statistical significance are reported.

Four different variants of empirical models are estimated. Model 1 has only the indicator variable for female-owned firms as an exogenous variable; Model 2 adds a set of country dummy variables, Model 3 adds a set of sector dummy variables, and Model 4 includes all control variables detailed in Section 2, too. Results are reported in Table 3.

**Table 3.** Gender of firm owner and firm exit in 10 European countries, 2019/2020: results from econometric models. Method: Probit (Average Marginal Effects); Dependent variable: Firm exit (1 = yes).

| Model Variable | | 1 | 2 | 3 | 4 |
|---|---|---|---|---|---|
| Female-owned firm | Average marginal effect | −0.00962 | −0.00885 | −0.00873 | −0.00928 |
| (Dummy; 1 = yes) | *p*-value | 0.078 | 0.106 | 0.112 | 0.088 |
| Web-presence | Average marginal effect | | | | −0.0252 |
| (Dummy; 1 = yes) | *p*-value | | | | 0.000 |
| Firm age | Average marginal effect | | | | −0.00067 |
| (Years) | *p*-value | | | | 0.003 |
| Firm size | Average marginal effect | | | | −0.01163 |
| (Log Number of employees) | *p*-value | | | | 0.000 |
| Exporter | Average marginal effect | | | | −0.0139 |
| (Dummy; 1 = yes) | *p*-value | | | | 0.042 |
| Foreign owned firm | Average marginal effect | | | | 0.0163 |
| (Dummy; 1 = yes) | *p*-value | | | | 0.286 |
| Product innovator | Average marginal effect | | | | −0.0114 |
| (Dummy; 1 = yes) | *p*-value | | | | 0.090 |
| Process innovator | Average marginal effect | | | | −0.0210 |
| (Dummy; 1 = yes) | *p*-value | | | | 0.006 |
| Country dummy variables | | no | yes | yes | yes |
| Sector dummy variables | | no | no | yes | yes |
| Number of observations | | 6013 | 6013 | 6013 | 6013 |

Source: Own calculations with data from World Bank Enterprise surveys; for details see text.

The most important result is that the estimated average marginal effect of being a female-owned firm, on firm exit, is negative and statistically significant at a ten percent level in all four empirical models. Irrespective of the control variables included, the model of female ownership in 2019 reduces the probability of firm exit until 2020.

With regard to the control variables included in Model 4, all of the estimated average marginal effects have the theoretically expected sign (as discussed in Section 2 above), and are statistically different from zero at an error level of 8 percent or much better, the only exception being the indicator for a foreign owned firm (where no clear theoretical hypothesis is found in the literature according to the discussion in Section 2 above).

Note that the estimated average marginal effect of female ownership on the chance to survive is about constant over all four models, so adding control variables does not change the results much. Furthermore, the size of this estimated effect can be considered as being large on average—the estimated average reduction in the probability of exit is slightly less than one percentage point, and this is really large compared with the overall exit probability of 4.59 percent in the sample reported in Table 1. Having a female owner helped firms to survive the negative shocks during the pandemic.

## 4. Concluding Remarks

This paper demonstrates that having a female owner is positively related to the probability of survival for firms facing negative demand and supply shocks during the COVID-19 pandemic. The estimated effect is statistically significant at a reasonable level ceteris paribus after controlling for various firm characteristics that are known to be positively related to survival. Furthermore, the size of this estimated effect can be considered as being large on average. Female owners helped firms to survive.

This finding might be explained by different degrees of risk aversion by gender. If women tend to be more risk averse than men, and firms led by men are engaged in more risky businesses, it follows that they will have a higher risk of failure, especially in times of negative demand and supply shocks; however, the data at hand are not rich enough to investigate whether this is the case. The empirical findings presented here are nevertheless interesting on its own.

**Funding:** This research received no external funding.

**Institutional Review Board Statement:** Not applicable.

**Informed Consent Statement:** Not applicable.

**Data Availability Statement:** The data presented in this study are available in this article.

**Acknowledgments:** I thank four anonymous referees for helpful comments on an earlier version. The data from the World Bank Enterprise surveys are available after registration from the website https://www.enterprisesurveys.org/portal/login.aspx, accessed on 20 February 2022. Stata code used to produce the empirical results reported in this note is available from the author.

**Conflicts of Interest:** The author declares no conflict of interest.

## Appendix A

**Table A1.** Descriptive statistics for sample used in estimations.

| Variable | Mean | Std. Dev. |
| --- | --- | --- |
| Female-owned firm (Dummy; 1 = yes) | 0.363 | 0.481 |
| Firm exit (Dummy; 1 = yes) | 0.046 | 0.209 |
| Web-presence (Dummy; 1 = yes) | 0.720 | 0.449 |
| Firm age (Years) | 21.05 | 15.28 |
| Firm size (Number of employees) | 81.64 | 354.73 |
| Product innovator (Dummy; 1 = yes) | 0.220 | 0.414 |
| Process innovator | 0.127 | 0.333 |

**Table A1.** *Cont.*

| Variable | Mean | Std. Dev. |
|---|---|---|
| (Dummy; 1 = yes) | | |
| Foreign owned firm (Dummy; 1 = yes) | 0.070 | 0.255 |
| Exporter (Dummy; 1 = yes) | 0.256 | 0.436 |
| Manufacturing (Dummy; 1 = yes) | 0.631 | 0.482 |
| Retail/Wholesale (Dummy; 1 = yes) | 0.201 | 0.401 |
| Construction (Dummy; 1 = yes) | 0.055 | 0.228 |
| Hotel/Restaurant (Dummy; 1 = yes) | 0.035 | 0.184 |
| Services (Dummy; 1 = yes) | 0.077 | 0.267 |

## Notes

[1]    The data from the World Bank Enterprise surveys are available free of charge after registration from the website https://www.enterprisesurveys.org/portal/login.aspx, accessed on 20 February 2022.

[2]    Not included are Albania, Cyprus, Estonia, Greece, Latvia, Lithuania, Malta, and Slovenia.

[3]    The questionnaires of the regular 2019 survey and the follow-up survey conducted in 2020 are available from the World Bank's Enterprise Survey web site referred to above.

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
