# Peer review of "Firm Survival and Gender of Firm Owner in Times of COVID-19: Evidence from 10 European Countries"

_economies, doi:10.3390/economies10050098_

Round 1

Reviewer 1 Report

Thank you very much for inviting me to review the paper entitled “Firm survival and gender of firm owner in times of COVID-19: Evidence from 10 European countries”.  By using firm level data from the World Bank Enterprise surveys, this paper tries to understand the link between the gender of the firm’s owner and firm survival until 2020. In this regards they found that female ownership is positive ceteris paribus after controlling for various firm characteristics that are known to be related to survival. Interestingly they found that having a female owner helped firms to survive. Overall, it’s an interesting research topic. I hope my comments help you to improve the paper and make it publishable.

In introduction section, there is a need to add at least one paragraph about the impact of gender on firms’ outcomes. The author(s) we defined the research gap (impact of owners’ gender on firms’ survival during health crisis). However, they stopped suddenly, and they didn’t explain well why filling this research gap is important first. And most importantly, why female ownership is going to help firm survival. Therefore, before talking about paper contribution to the literature, I strongly recommend author(s) to add one paragraph and clearly explain their reasons for impact of gender (being a women owner) on firm survival.

My second recommendation to the author(s) is about avoiding ambiguous words in your paper.  For instance, you mention that “Some firms were hit so hard by these negative exogenous shocks that they decided 21 to close down permanently. An important question is which characteristics of firms help 22 many of them to survive the pandemic”. Instead of many firms, do you have a percentage of failed firms, or percentage of successful firms during covid pandemic?

My third and lost concern is about your control variables. You have considered many relevant factors as control variable, I appreciate it. As you mentioned here: “In the empirical investigation of the link between gender of the firm’s owner and firm survival a number of firm characteristics that are known to be correlated with firm exit (and that might be related to the gender of the firm’s owner as well) are controlled for.” If I am not wrong, you missed one important control variable here: firms’ industrial affiliation.  We know that many companies especially in healthcare sectors benefits a lot for this ongoing health crisis. Or businesses in entertainment sector, e-learning, remote working tools and software, pharmaceutical and medical devices, and logistics sectors boosted by coronavirus concerns.

Author Response

Comments on the report of Referee 1

I thank the referee for carefully reading my paper and making suggestions to improve it. Below my comments to each point raised by the referee are included in red.

Thank you very much for inviting me to review the paper entitled “Firm survival and gender of firm owner in times of COVID-19: Evidence from 10 European countries”.  By using firm level data from the World Bank Enterprise surveys, this paper tries to understand the link between the gender of the firm’s owner and firm survival until 2020. In this regards they found that female ownership is positive ceteris paribus after controlling for various firm characteristics that are known to be related to survival. Interestingly they found that having a female owner helped firms to survive. Overall, it’s an interesting research topic. I hope my comments help you to improve the paper and make it publishable.

In introduction section, there is a need to add at least one paragraph about the impact of gender on firms’ outcomes. The author(s) we defined the research gap (impact of owners’ gender on firms’ survival during health crisis). However, they stopped suddenly, and they didn’t explain well why filling this research gap is important first. And most importantly, why female ownership is going to help firm survival. Therefore, before talking about paper contribution to the literature, I strongly recommend author(s) to add one paragraph and clearly explain their reasons for impact of gender (being a women owner) on firm survival.

I agree. I added one paragraph in the motivation section to do so.

My second recommendation to the author(s) is about avoiding ambiguous words in your paper.  For instance, you mention that “Some firms were hit so hard by these negative exogenous shocks that they decided 21 to close down permanently. An important question is which characteristics of firms help 22 many of them to survive the pandemic”. Instead of many firms, do you have a percentage of failed firms, or percentage of successful firms during covid pandemic?

I agree. I added this information in the second paragraph of the introductory section.

My third and lost concern is about your control variables. You have considered many relevant factors as control variable, I appreciate it. As you mentioned here: “In the empirical investigation of the link between gender of the firm’s owner and firm survival a number of firm characteristics that are known to be correlated with firm exit (and that might be related to the gender of the firm’s owner as well) are controlled for.” If I am not wrong, you missed one important control variable here: firms’ industrial affiliation.  We know that many companies especially in healthcare sectors benefits a lot for this ongoing health crisis. Or businesses in entertainment sector, e-learning, remote working tools and software, pharmaceutical and medical devices, and logistics sectors boosted by coronavirus concerns.

I agree that detailed industry controls would be very informative. However, the data at hand are not rich enough to include such controls. In the empirical models I control for sector dummy variables (Manufacturing, Retail/Wholesale, Construction, Hotel/Restaurant, Services) using the information available in the data.

Reviewer 2 Report

1-Abstract section might include:

Your research problem and objectives

Your methods

Your key results or arguments or limitation

Your conclusion

2-Relationship to Literature:

Develop the literature review. The paper presents Firm survival and gender of firm owner in times of COVID-19 However, the practices related to 10 European countries outcomes not discussed in deepen and why they were chosen.

3-Methodology:  The questions are weakly built. There is no previous theoretical discussion to propose them

4-Results:  presented clearly and analysed appropriately, they are presented with clarity.

5-Theoretical implications are limited considering the problems mentioned.  Considering applied implications, there are no findings regarding regional issues as proposed by the authors as the paper objective.

6-The quality of communication is satisfactory.

7-The lack of a proper objective needs and purpose for the paper are the main weakness.

8-It seems to me the references are out-dated. 10 % of the references are from the past four years

9- Too many citation (Wagner)

10- Develop the literature review part of the paper to include 6 to 8 latest journal references (2018.2022)

I hope you find the comments and suggestions important to develop your paper for publication. Overcoming them will improve its quality.  

Author Response

Comments on the report of Referee 2

I thank the referee for carefully reading my paper and making suggestions to improve it. Below my comments to each point are included in red.

1-Abstract section might include:

Your research problem and objectives

Your methods

Your key results or arguments or limitation

Your conclusion

I added a sentence on the method used. All other information is in the abstract.

2-Relationship to Literature:

Develop the literature review. The paper presents Firm survival and gender of firm owner in times of COVID-19 However, the practices related to 10 European countries outcomes not discussed in deepen and why they were chosen.

The selection of countries is discussed at the beginning of section 2. The data, however, are not rich enough to discuss any differences by country – the samples for the individual countries are not comprehensive enough for such an exercise. Furthermore, this is a short note that has a focus on one (in my view, interesting) new result on a topic that has not been investigated before, so I do not think that a broad literature review is necessary.

3-Methodology:  The questions are weakly built. There is no previous theoretical discussion to propose them

I added one paragraph in the introductory section on this.

4-Results:  presented clearly and analysed appropriately, they are presented with clarity.

Thank you.

5-Theoretical implications are limited considering the problems mentioned.  Considering applied implications, there are no findings regarding regional issues as proposed by the authors as the paper objective.

Regional issues are beyond the scope of this note – see my remarks on point 2.

6-The quality of communication is satisfactory.

Than you.

7-The lack of a proper objective needs and purpose for the paper are the main weakness.

I hope the extension to the introduction helps to overcome this critique.

8-It seems to me the references are out-dated. 10 % of the references are from the past four years

The references are to the “classical” sources that discuss the control variables and that are standard.

9- Too many citation (Wagner)

I think all these references are necessary.

10- Develop the literature review part of the paper to include 6 to 8 latest journal references (2018.2022)

 As said, this is a short note on a new topic that has not been investigated empirically before. There are to the best of my knowledge no recent papers to discuss.

I hope you find the comments and suggestions important to develop your paper for publication. Overcoming them will improve its quality.

Reviewer 3 Report

This paper studies how the gender composition of a firm’s owners affects the survival of this firm. Using World Bank Enterprise surveys data, this paper show that there is positive correlation between the female-owned firm in 2019 and the chance of survival in 2020. The gender of a firm’s owner and their reaction under the negative shock is an interesting question to study. However, I have several major concerns about this paper:

Major concerns:

  1. All the tables in this paper are extremely difficult to read in terms of their format. Without a proper design of these tables, I cannot fully understand these numbers and the corresponding conclusions.
  2. The variables used in this paper cannot accurately proxy for the concept of the research question:
  • Female-owned: this paper defines a firm is female-owned if there is at least one female amongst the owners of the firm. Without knowing fraction of females of the firms’ owners or the equity shares owned by females, we do not know the females’ influence on firms’ decision making.
  • Survival: this paper defines the survey answer “permanently closed” as firms that failed to survive in this paper. In this definition, if a firm is acquired by another firm, it failed to survive. It is better to clarify the reasons behind this “permanently closed”( e.g. financial distress, acquired, etc.)

Beside the major concerns, there are several issues about the sample and research design of this paper:

Data sample and Research Design

  1. The paper should list the firm characteristics for both the female-owned firms and non-female-owned firms and compare their differences at the first place. We can see how the firm-level characteristics are different between the two samples.
  2. This paper controls a bunch of dummy variables except two, firm size and firm age. Other important variables should not be neglected as to firms’ survival. For examples, the profitability, leverage, and other financial information of these firms.
  3. For firm size, you should take logarithm instead of use the raw number.

Author Response

Comments on the report of Referee 3

I thank the referee for carefully reading my paper and making suggestions to improve it. Below my comments to each point raised by the referee are included in red.

This paper studies how the gender composition of a firm’s owners affects the survival of this firm. Using World Bank Enterprise surveys data, this paper show that there is positive correlation between the female-owned firm in 2019 and the chance of survival in 2020. The gender of a firm’s owner and their reaction under the negative shock is an interesting question to study. However, I have several major concerns about this paper:

Major concerns:

  1. All the tables in this paper are extremely difficult to read in terms of their format. Without a proper design of these tables, I cannot fully understand these numbers and the corresponding conclusions.

Frankly, I have no idea what your concerns are here. I use a standard layout for the tables that is used in many papers before – and I never was told that my tables are extremely difficult to read.

  1. The variables used in this paper cannot accurately proxy for the concept of the research question:
  • Female-owned: this paper defines a firm is female-owned if there is at least one female amongst the owners of the firm. Without knowing fraction of females of the firms’ owners or the equity shares owned by females, we do not know the females’ influence on firms’ decision making.

I agree that more information would be fine here. Unfortunately, it is not available to me.

  • Survival: this paper defines the survey answer “permanently closed” as firms that failed to survive in this paper. In this definition, if a firm is acquired by another firm, it failed to survive. It is better to clarify the reasons behind this “permanently closed”( e.g. financial distress, acquired, etc.)

As far as I understand, a firm that is acquired by another firm and that is still located at the old place is not considered to be permanently closed. Further information on the reason why a firm classified as “permanently closed”, however, is not available in the data.

Beside the major concerns, there are several issues about the sample and research design of this paper:

Data sample and Research Design

  1. The paper should list the firm characteristics for both the female-owned firms and non-female-owned firms and compare their differences at the first place. We can see how the firm-level characteristics are different between the two samples.

I agree. I added a new table 2 with this information and include some vomments at the end of section 2.

  1. This paper controls a bunch of dummy variables except two, firm size and firm age. Other important variables should not be neglected as to firms’ survival. For examples, the profitability, leverage, and other financial information of these firms.

I agree. However, this information is not available in the data.

  1. For firm size, you should take logarithm instead of use the raw number.

I agree. Firm size is now in logs – see the (new) table 3.

Reviewer 4 Report

Thank you for submitting your paper. The article provides an interesting analysis of the link between the gender of the firm’s owner and firm survival until 2020.

Comments on individual parts:

  • The abstract is not precise enough. The abstract needs to address the motivations, contributions and results of the work, concisely, clearly and persuasively.
  • The Introduction highlights the research problem and defines the research gaps. However, it is necessary to review the literature on the topic, and noting studies that have used similar analysis. Note how your study clarifies existing knowledge by utilizing more literature.

Moreover, the introduction should provide not only a research background but also a research objective and a brief indication of the methods used.

  • The literature review is missing in the paper. The literature review should present important highlights on the state of the art by utilizing more literature from the leading journals.
  • The use of research methods is adequate. However, the choice of research method should be better justified.
  • The result section includes data analysis. The key findings can be considered significant and important. However, the result section should include not only data presentation but also the interpretation of the findings obtained. The key findings presented in the table should be better explained and justified considering more details. The critical assessment of the results will improve the quality of the analysis.
  • The discussion section is missing. The discussion should meld together your findings in relation to those identified in the literature review, and be placed within the context of the theoretical framework underpinning the study. It should embrace an extensive explanation of differences and similarities between the findings obtained and those of other scholars. Moreover, please describe how the results helped to fill the gap in understanding the research problem?
  • The conclusions section includes a synthetic overview of the key research results. However, the authors should also indicate practical or/and theoretical implications, research limitations and potential directions for further research.
  • Furthermore, it is necessary to refer to more recent literature. The list of references includes only 17 literature sources. It would be valuable if you could use the more recent literature.

The academic language is correct, but general proofreading would be advisable.

Author Response

Comments on the report of Referee 4

I thank the referee for carefully reading my paper and making suggestions to improve it. Below my comments to each point raised by the referee are included in red.

Thank you for submitting your paper. The article provides an interesting analysis of the link between the gender of the firm’s owner and firm survival until 2020.

Comments on individual parts:

  • The abstract is not precise enough. The abstract needs to address the motivations, contributions and results of the work, concisely, clearly and persuasively.

I hope that the (revised) abstract does all this.

  • The Introduction highlights the research problem and defines the research gaps. However, it is necessary to review the literature on the topic, and noting studies that have used similar analysis. Note how your study clarifies existing knowledge by utilizing more literature.

This topic has – to the best of knowledge – not been discussed in the literature beforeand there are no other studies to discuss here.

  • Moreover, the introduction should provide not only a research background but also a research objective and a brief indication of the methods used.

This is covered in the motivation section.

  • The literature review is missing in the paper. The literature review should present important highlights on the state of the art by utilizing more literature from the leading journals.

This paper is intended to be a short note on an (in my view) important topic that present a ne new and interesting result. It is not intended to be a full-length paper on firm survival as such. Therefore, in my view it suffices to refer to the survey paper by Jennings and Brush for the broader literature on gender and firm performance, and to the classical references that motivate the control variables.

  • The use of research methods is adequate. However, the choice of research method should be better justified.

The empirical method is standard in this area of research and I do not see how it should be better justified.

  • The result section includes data analysis. The key findings can be considered significant and important. However, the result section should include not only data presentation but also the interpretation of the findings obtained. The key findings presented in the table should be better explained and justified considering more details. The critical assessment of the results will improve the quality of the analysis.

Again, please note that this is a short note on a specific topic. The presentation and discussion concentrates on this topic.

  • The discussion section is missing. The discussion should meld together your findings in relation to those identified in the literature review, and be placed within the context of the theoretical framework underpinning the study. It should embrace an extensive explanation of differences and similarities between the findings obtained and those of other scholars. Moreover, please describe how the results helped to fill the gap in understanding the research problem?

To repeat, this is a short note on a new topic that has not been discussed in the literature before.

  • The conclusions section includes a synthetic overview of the key research results. However, the authors should also indicate practical or/and theoretical implications, research limitations and potential directions for further research.

I think this is done in the last paragraph of the  note.

  • Furthermore, it is necessary to refer to more recent literature. The list of references includes only 17 literature sources. It would be valuable if you could use the more recent literature.

To repeat, the topic of the note has not been discussed in the literature before, so there are no recent studies to be discussed. Most of the (partly somewhat dated) references are needed to justify the selection of the control variables used in the empirical models. 

The academic language is correct, but general proofreading would be advisable.

The paper has been checked carefully.

Round 2

Reviewer 2 Report

This manuscript ticks bout not all the boxes we have in mind for an " Firm survival and gender of firm owner in times of COVID-19 Evidence from 10 European countries" paper. I have no hesitation in recommending that it will be accepted for publication after a few typos and other Minor details have been attended to.

Given the complexity involved, the author has now produced many positive and welcome outcomes

Reviewer 3 Report

The paper has been improved after revision. But I can still see lots of strange symbols in the tables. I am not sure if they are supposed to be there.